# CRISPR/Cas9-mediated gene targeting in *Arabidopsis* using sequential transformation

Daisuke Miki[1], Wenxin Zhang[1,2], Wenjie Zeng[1,2], Zhengyan Feng[1] & Jian-Kang Zhu[1,3]

Homologous recombination-based gene targeting is a powerful tool for precise genome modification and has been widely used in organisms ranging from yeast to higher organisms such as *Drosophila* and mouse. However, gene targeting in higher plants, including the most widely used model plant *Arabidopsis thaliana*, remains challenging. Here we report a sequential transformation method for gene targeting in *Arabidopsis*. We find that parental lines expressing the bacterial endonuclease Cas9 from the egg cell- and early embryo-specific *DD45* gene promoter can improve the frequency of single-guide RNA-targeted gene knock-ins and sequence replacements via homologous recombination at several endogenous sites in the *Arabidopsis* genome. These heritable gene targeting can be identified by regular PCR. Our approach enables routine and fine manipulation of the *Arabidopsis* genome.

[1] Shanghai Center for Plant Stress Biology and Center for Excellence in Molecular Plant Sciences, Chinese Academy of Sciences, 200032 Shanghai, China. [2] University of Chinese Academy of Sciences, 100049 Beijing, China. [3] Department of Horticulture and Landscape Architecture, Purdue University, West Lafayette, IN 47907, USA. These authors contributed equally: Daisuke Miki, Wenxin Zhang. Correspondence and requests for materials should be addressed to D.M. (email: miki@sibs.ac.cn) or to J.-K.Z. (email: jkzhu@sibs.ac.cn)

Precise genome modification such as DNA knock-in and gene replacement (i.e., gene targeting) via homologous recombination is a powerful tool that is widely applied for research in many organisms, including *Drosophila* and animals[1–3]. However, gene targeting (GT) is still very challenging in higher plant species, because of low efficiency of homologous recombination[4].

Engineered sequence-specific nucleases such as zinc finger nucleases (ZFNs), transcription activator-like effector nucleases (TALENs) and clustered regularly interspaced short palindromic repeats (CRISPR)/CRISPR-associated protein 9 (Cas9) have been used to generate site-specific double stranded breaks (DSBs) for genome editing in numerous organisms[1,5–7]. Repair of these DSBs via error-prone non-homologous end-joining (NHEJ) leads to random mutations, whereas error-free homology-directed repair (HDR) creates precise sequence changes when a homologous DNA substrate is provided. A goal of genome editing is to achieve heritable GT, defined as the precise insertion or replacement of sequence at any genomic locus of interest in germline cells.

However, HDR-mediated GT at endogenous genes is extremely inefficient in higher plants, preventing its widespread application[4]. The first GT in plants was demonstrated at a kanamycin resistance gene in tobacco, with a frequency ranging from $10^{-3}$ to $10^{-6}$ (refs. [8,9]). A higher efficiency method using positive −negative selection was later developed in rice[10]; however, this complicated strategy has been used to modify only several genes in rice[11] and has not been successfully applied to other plants, including *Arabidopsis*[12,13] and tobacco[14]. Sequence-specific nucleases can increase the efficiency of GT[1,15,16], and CRISPR/Cas9-assisted HDR has been used for GT in various model systems, including human stem cells[15]. The introduction of DSBs also increased the frequency of HDR in plants[17,18], and recent publications report using sequence-specific nucleases for HDR-mediated GT in *Arabidopsis*[19–24], tobacco[25–30], soybean[31], tomato[32,33], rice[34–41], maize[42–46], wheat[47,48], potato[49], barley[50], flax[51], and cotton[52]. Nevertheless, these GT events mostly relied on selection for antibiotic or herbicide resistance genes at the targeted loci to improve efficiency. The few GT events that did not rely on selection markers displayed extremely low frequencies[24,31,43], thus limiting the usefulness of these methods.

Here, we describe a simple method for seamless GT in *Arabidopsis*, including in-frame gene knock-ins and amino acid substitutions. We demonstrate the utility of our method by targeting the endogenous DNA glycosylase genes *ROS1* and *DME* in *Arabidopsis*.

## Results

**Inefficient GT by an all-in-one strategy**. To achieve efficient GT in *Arabidopsis*, we first designed an "all-in-one" T-DNA construct that contains: (i) Cas9 driven by the CaMV 35S promoter (35Spro::Cas9), (ii) an sgRNA driven by the AtU6 promoter, that targets a site near the stop codon of *ROS1*, and (iii) a donor DNA fragment for in-frame *GFP* knock-in (Supplementary Fig. 1a). We screened T1 plants by PCR (e.g. Supplementary Fig. 1b), and identified 2/30 with a positive GT signal (Supplementary Table 1). In contrast, a control construct without an sgRNA did not yield any T1 plants with a positive GT signal (Supplementary Table 1). Neither of the T1-positive plants gave rise to T2 progenies with a positive GT signal, although bulk screening of 18 remaining T2 lines identified a positive GT signal (Supplementary Table 1). Southern blot analysis of individual plants from this PCR-positive T2 population failed to detect any GT-positive plants (Supplementary Fig. 1b), suggesting that the GT-positive PCR signal may have come from a small number of somatic cells. Thus, this

method did not generate heritable GT. A similar all-in-one construct also failed to generate heritable in-frame *ROS1-Luc* knock-ins (Supplementary Table 1).

The expression of Cas9 under germline-specific promoters was recently shown to increase the efficiency of CRISPR/Cas9-mediated gene editing in *Arabidopsis*[53–55]. We hypothesized that driving Cas9 expression from a germline-specific promoter instead of the CaMV 35S promoter might increase the frequency of heritable GT. We tested the following promoters: the egg cell- and early embryo-specific promoter DD45[53,54,56], the pollen-specific promoter Lat52[53], and the shoot apical meristem-active promoters YAO[55] and CDC45[57]. We generated all-in-one constructs for *GFP* knock-in into the *GLABRA2* (*GL2*) locus, utilizing these promoters to drive Cas9 expression and an sgRNA known to efficiently generate site-specific DSB in *GL2*[53] (Supplementary Fig. 1c). Although we observed high frequencies of GT-positive PCR signals with some of these all-in-one constructs, we did not identify any heritable GT lines (Supplementary Fig. 1d–i, Supplementary Table 2). Sequencing of the PCR products indicated that precise GT events occurred, but they likely represent minor events in some somatic cells. Thus, although expression of Cas9 under these specific promoters might improve GT efficiency in some somatic tissues, it did not lead to heritable GT.

**Knock-in into the *ROS1* locus by sequential transformation**. Next, we used a "sequential transformation method" to evaluate GT efficiency[35,41] in parental *Arabidopsis* plants that already express Cas9 from a germline-specific (DD45, Lat52, YAO or CDC45) promoter (Fig. 1). These parental Cas9 lines also express a *GL2*-targeting sgRNA from the AtU6 promoter. We used the two highest efficiency CRISPR/Cas9 lines, which were screened from 32 to 36 independent T1 lines based on the mutation rates at the *GL2* locus, for each specific promoter[53]. We used these Cas9-expressing plants as parental lines for new transformations with a construct containing: (i) HDR donor sequence, (ii) sgRNA targeting a genomic locus of interest, (iii) a selectable marker for plants that are positive for the donor construct (Figs. 1, 2a). The new transformation T1 transgenic plants were selected using the Basta resistance gene. These T1 plants express Cas9 and a specific sgRNA, and contain a specific HDR donor sequence. T1 seeds were harvested and germinated without selection on MS plates; 20−30 of the resulting T2 seedlings were subsequently pooled together, and GT events were analyzed by PCR in bulk. Further, another batch of T2 plants from the bulk positive lines were investigated as individual plants (Fig. 1).

Transformation of a construct containing *ROS1*-targeting sgRNA and *ROS1-GFP* donor sequence into DD45pro::Cas9 lines #58 and #70, but not other promoter::Cas9 lines, gave rise to Southern blot- and PCR-positive GT signals (Fig. 2a−c, Table 1, Supplementary Fig. 2, Supplementary Table 3). Six out of 11 tested plants from two T2 populations in the DD45-#58 background were homozygous *ROS1-GFP* GT lines based on Southern blot analysis, and 2 of 12 tested plants from another two T2 populations in the DD45-#70 background were homozygous (Table 1; e.g. Fig. 2c). Sanger sequencing confirmed that there were no mutations in the 5′ and 3′ homology arms and their border regions, and that *GFP* integration downstream of the *ROS1* gene was in-frame (Supplementary Figs. 4a and 5a). We examined the progenies of a heterozygous T2 GT plant and found that the integrated *ROS1-GFP* segregated in T3 (Fig. 2d). We analyzed mRNA expression in these T3 plant samples by RT-PCR and qRT-PCR, and observed comparable expression of the *ROS1-GFP* knock-in with endogenous *ROS1* (Fig. 2e, f). Further, the root tissues of homozygous T3 *ROS1-GFP* plants displayed GFP

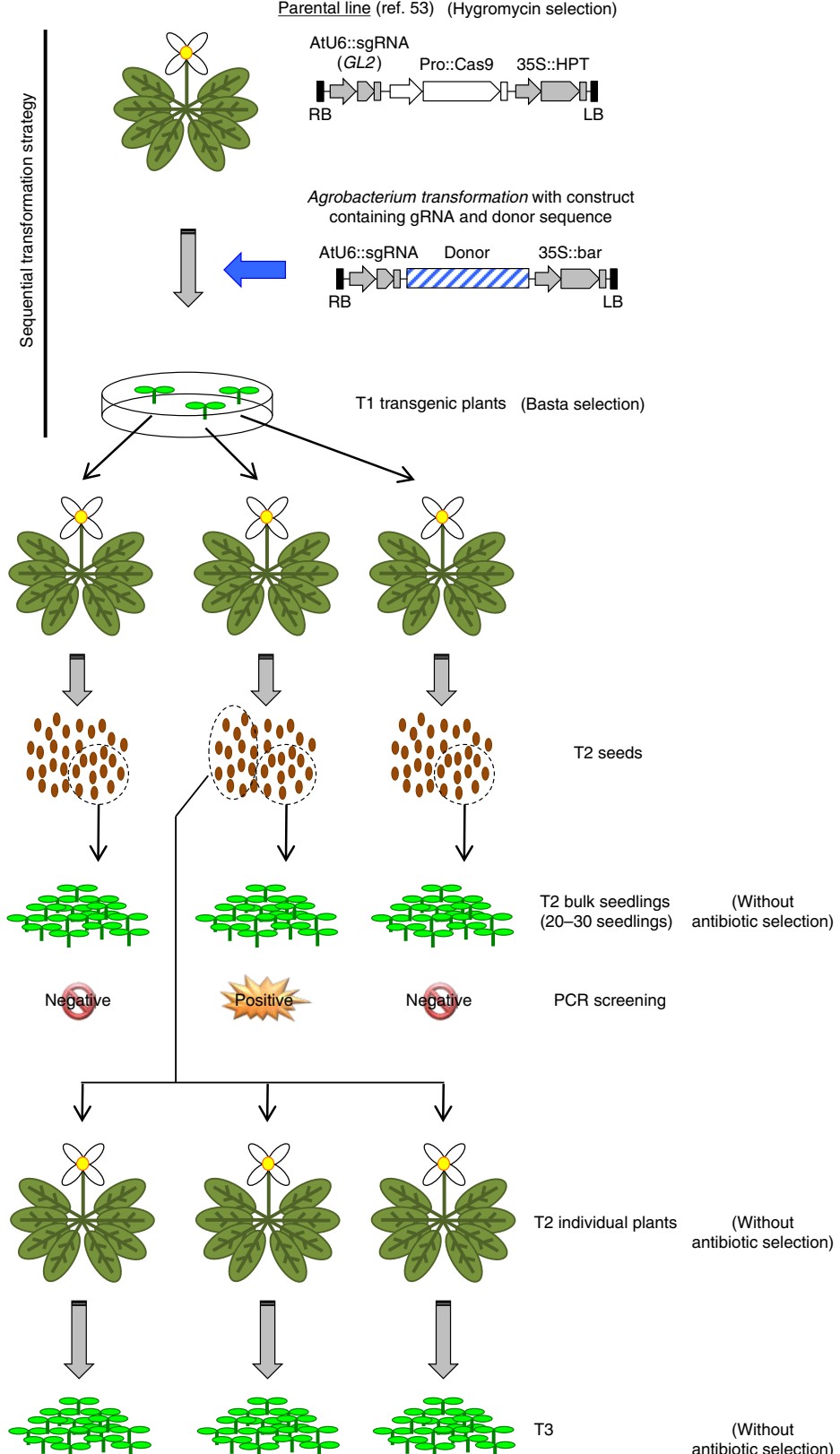

**Fig. 1** Outline of the sequential transformation strategy for gene targeting and screening procedure. HDR donor constructs containing a selection marker and sgRNAs targeting genes of interest were transformed into parental lines, which were already transgenic for an incidental *GL2*-targeting sgRNA and, importantly, for Cas9 driven by a specific promoter. T1 transgenic lines were selected with Basta, and T2 seedlings were obtained and analyzed in bulk. The positive lines were used for further experiments with individual T2 plants. Portions of the images were obtained from the Microsoft PowerPoint clip art data base

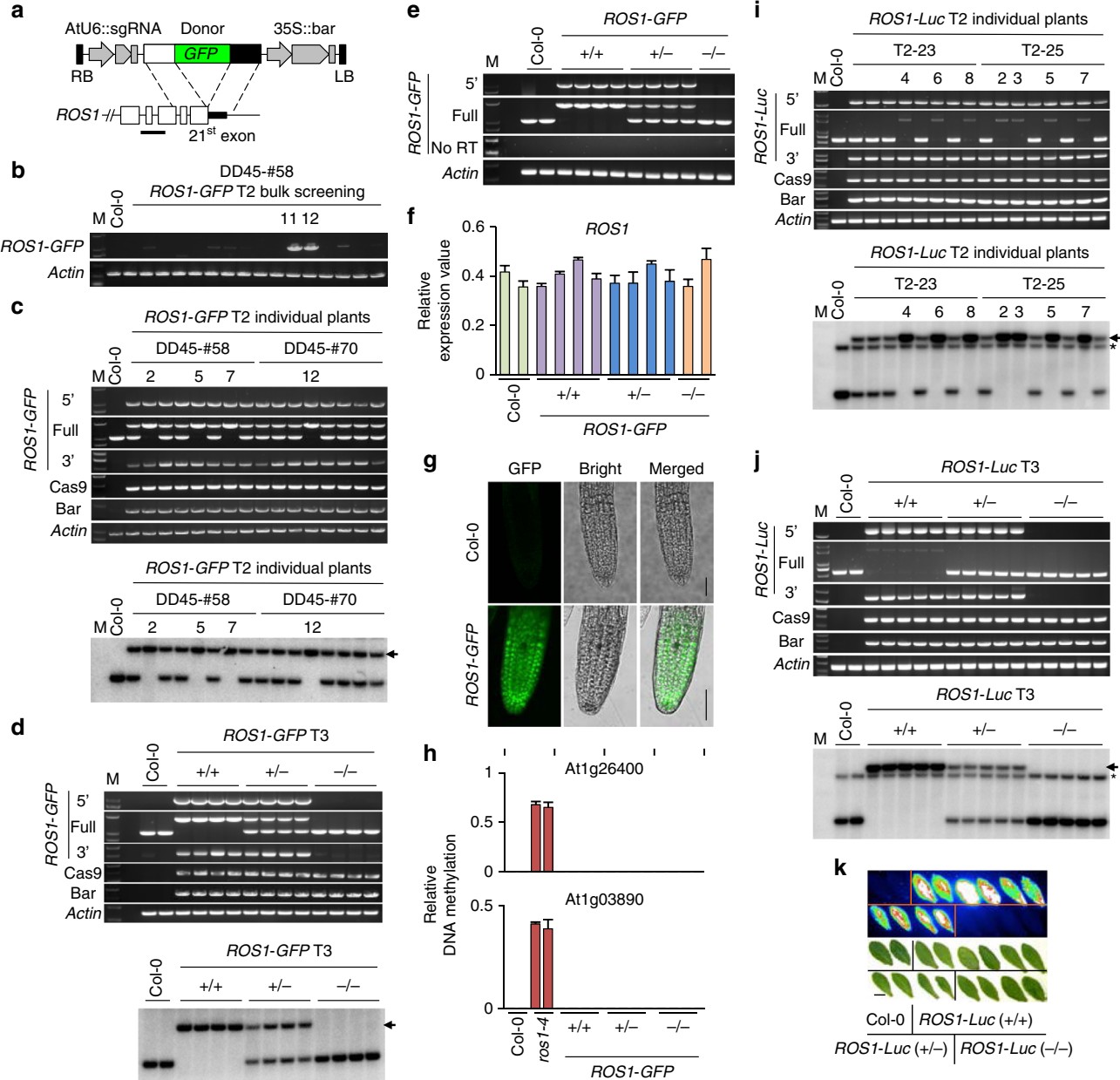

**Fig. 2** *GFP* integration into the endogenous *ROS1* locus by gene targeting. **a** Schematics showing HDR donor transgene construct (top) and part of the targeted *ROS1* locus (bottom). The horizontal line indicates the position of a probe for Southern blot hybridization. **b** PCR screening of T2 bulk samples (see Supplementary Figure 2). Primers for the 5′ part of *ROS1-GFP* were used (see Supplementary Figure 2). **c, d** Genotyping PCR and Southern blotting of a subset of individual T2 (**c**) and T3 plants (**d**) of *ROS1-GFP*, respectively. The eight plants in the DD45-#58 background were selected from populations 11 and 12 in (**b**). Arrow indicates the band of *ROS1-GFP* from gene targeting. **e** RT-PCR and **f** qRT-PCR on the same T3 plants as in (**d**). The error bars indicate *P* values of Student's *t* test (*n* = 4). See Supplementary Figure 2. **g** Detection of GFP fluorescence. Scale bar, 50 μm. **h** qChop-PCR at the At1g26400 and At1g03890 loci. The *ros1-4* mutant was used as a positive control. The error bars indicate *P* values of Student's *t* test (*n* = 4). **i, j** Genotyping PCR and Southern blotting for individual T2 lines (**i**) and T3 plants (**j**) of *ROS1-Luc*, respectively. Arrow indicates the band of *ROS1-Luc* from gene targeting, and the asterisk denotes non-specific cross-hybridization band. **k** Luminescence in T3 *ROS1-Luc* leaves. Scale bar, 1 cm. All PCR primers are as depicted in Supplementary Figure 2b, c and Supplementary Table 4

fluorescence (Fig. 2g). To determine whether the *ROS1-GFP* knock-in retained ROS1 function, we assessed the DNA methylation level of two genomic loci known to become hypermethylated in loss-of-function *ros1* mutant plants by quantitative Chop-PCR (Fig. 2h)[58]. Homozygous T3 *ROS1-GFP* knock-in plants did not display hypermethylation at these loci, suggesting that the in-frame integration of *GFP* did not interfere with *ROS1* function, and that the *ROS1-GFP* was functional.

Thus, our sequential transformation method efficiently generates precise and heritable GT.

Next we tested whether a fragment longer than *GFP* could be integrated at the *ROS1* locus. We used the same sgRNA and homology arms to make a donor construct that contained firefly luciferase (*Luc*: 1653 bp) instead of *GFP* (720 bp), and transformed the construct into parental CRISPR/Cas9 lines. Two positive GT lines were identified in T2 bulk screening by

**Table 1 Knock-in GT efficiencies for the _ROS1_ and _DME_ loci**

| Construct | Parental line | T2 | | | | | | GT efficiency |
|---|---|---|---|---|---|---|---|---|
| | | Bulk | | Individual plants | | | | |
| | | PCR | Line name | PCR | Southern | Homo | | |
| _ROS1-GFP_ | DD45-#58 | 2/26 | T2-11 | 7/56 | 7/7 | 4 | | 7.7% (2/26) |
| | | | T2-12 | 4/59 | 4/4 | 2 | | |
| | DD45-#70 | 2/24 | T2-6 | 8/66 | 8/8 | 2 | | 8.3% (2/24) |
| | | | T2-11 | 10/62 | 4/4 | 0 | | |
| _ROS1-Luc_ | DD45-#58 | 2/32 | T2-23 | 20/65 | 20/20 | 3 | | 6.3% (2/32) |
| | | | T2-25 | 10/72 | 10/10 | 4 | | |
| _DME-GFP_ | DD45-#58 | 2/22 | T2-9 | 20/60 | 20/20 | 16 | | 9.1% (2/22) |
| | | | T2-14 | 42/60 | 42/42 | 15 | | |
| _GFP-DME_ | DD45-#58 | 2/24 | T2-11 | 4/57 | 4/4 | 1 | | 8.3% (2/24) |
| | | | T2-24 | 11/60 | 6/6 | 3 | | |
| | DD45-#70 | 2/24 | T2-2 | 18/54 | 18/18 | 6 | | 8.3% (2/24) |
| | | | T2-13 | 7/64 | 7/7 | 0 | | |

The T2 bulk positive populations were subjected to PCR analysis using T2 individual plants, and the positive individuals were then analyzed by Southern blotting. GT efficiency was calculated based on the number of T2 populations examined

PCR, and precise knock-in was confirmed in individual T2 and T3 plants (Table 1, Figs. 1, 2i, j). These true GT-positive (PCR and Southern blotting positive in individual T2 plants) _ROS1-Luc_ lines were all from the DD45pro::Cas9 background (Table 1, Supplementary Table 3). The leaves of homozygous and heterozygous _ROS1-Luc_ T3 plants displayed luminescence signals, unlike those from control plants without GT (Fig. 2k). Thus, a fragment as large as 1.6 kb can be stably integrated into a genomic locus using our sequential transformation GT strategy.

**Knock-in into the _DME_ locus**. Next, to investigate the broad utility of our GT method, we attempted to generate in-frame _GFP_ knock-ins at the 5′ end and the 3′ end of _DME_ (At5g04560), a DNA glycosylase gene on a different chromosome than _ROS1_ in _Arabidopsis_. We designed specific sgRNAs and donor constructs for a 3′ in-frame fusion (_DME-GFP_) and 5′ in-frame fusion (_GFP-DME_) (Fig. 3a, b, Supplementary Fig. 3). The sgRNA used to generate _GFP-DME_ also targets the 3′ homology region of the donor construct, so we introduced silent mutations within the 3′ donor sequence of _GFP-DME_ to prevent sgRNA binding, DSB and mutations following precise knock-in (Supplementary Fig. 3b).

These T-DNA constructs were transformed into the parental Lat52, YAO, CDC45, and DD45 promoter-driven CRISPR/Cas9 lines. Although some GT signals were detected by PCR in the T1 and T2 plants from the Lat52, YAO and CDC45 parental lines, they were not heritable GT events, given that positive signals were not detected by Southern blotting or in some cases even by PCR in individual T2 plants (Supplementary Table 3). In contrast, true GT-positive (PCR and Southern blotting positive in individual T2 plants) signals were detected for _DME-GFP_ from 2 out of 22 T2 populations (9.1%) in the DD45-#58 parental line (Table 1). Further, two positive GT signals were detected for _GFP-DME_ from 24 T2 populations (8.3%), from each of the DD45-#58 and DD45-#70 parental lines (Table 1). Analysis of individual T2 plants revealed homozygous and heterozygous plants for both _DME-GFP_ and _GFP-DME_ fusions (Fig. 3c, d, Supplementary Fig. 3). The heterozygous T2 plants segregated in T3 (Fig. 3e, f, Supplementary Fig. 3). These in-frame _GFP_ knock-ins at the 5′ and 3′ ends of _DME_ were confirmed by sequencing the PCR products (Supplementary Figs. 4b, c, 5b, c).

Homozygous and heterozygous _DME-GFP_ and _GFP-DME_ plants did not show any developmental or growth defects, suggesting that the gene-targeted _DME_ is functional, since _dme_

loss-of-function mutants show maternal lethality[59]. To further confirm that the _DME-GFP_ and _GFP-DME_ in-frame fusion proteins are functional, we examined the seed abortion ratios of homozygous _DME-GFP_ and _GFP-DME_ T3 plants, and found that they were comparable with that of wild-type Col-0 plants (Fig. 3g). Thus, the _DME-GFP_ and _GFP-DME_ in-frame fusions are functional.

**Sequence replacement at the _DME_ locus**. An important goal of GT is the fine manipulation of endogenous genes by gene replacement. To test the feasibility of gene replacement, we attempted to substitute an amino acid within a conserved motif of _DME_ (Supplementary Fig. 6). The Fe-S motif is highly conserved in the family of 5-methylcytosine DNA glycosylases, and is required for 5-methylcytosine DNA glycosylase activity of _DME_ and _ROS1_ in vitro[60,61]. We generated mutated forms of a _DME_ donor by changing a conserved proline to alanine (P1633A) and phenylalanine to alanine (F1648A). Silent mutations were also integrated at the PAM sequence to block additional DSBs, following the CORRECT method[15] (Supplementary Fig. 6). The two constructs containing the mutated _DME_ donors and corresponding sgRNAs were transformed into YAO, CDC45, and DD45 promoter-driven CRISPR/Cas9 parental lines. We used a PCR-restriction enzyme assay to uncover amino acid substitution GT events. Heritable GT lines were obtained only in the DD45pro::Cas9 parental background (Fig. 4a, b, Table 2, Supplementary Table 3). We sequenced the PCR amplicons from GT-positive T2 plants and found accurate amino acid substitutions, with no other mutations (Fig. 4c, d). Southern blot analysis of several T3 plants revealed that they were all heterozygous for the amino acid substitution GT (Fig. 4e). Thus, the amino acid substitution GT was stable and heritable.

We did not obtain any homozygous _P1633A_ and _F1648A_ GT plants in T2 or T3 generations, likely due to the lethality of loss-of-function _dme_ mutations[59]. Indeed, approximately 50% of the seeds of the _P1633A_ and _F1648A_ heterozygous T3 plants aborted, whereas no seed abortion was found in T3 plants without the amino acid substitution GT (Fig. 4f). Thus, these two highly conserved amino acids within the Fe-S motif, P1633 and F1648, are essential for _DME_ function in vivo.

**GT effect on DNA methylation**. ZFN-mediated GT of the endogenous locus _PPOX_ in plants reportedly alters its epigenetic status[62]. We performed individual locus bisulfite

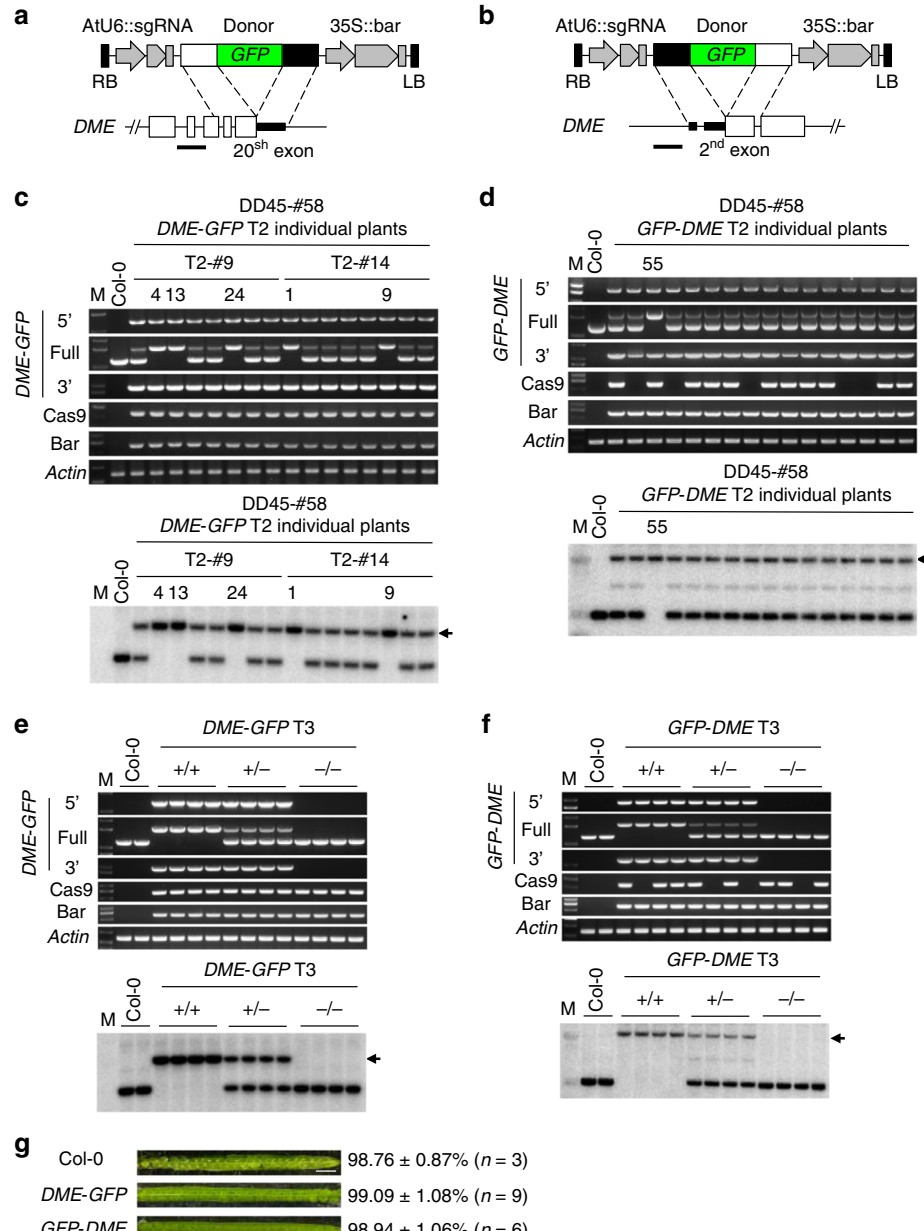

**Fig. 3** *GFP* knock-in into the endogenous *DME* locus by gene targeting. **a**, **b** Schematics showing HDR donor transgene constructs and part of the targeted *DME* locus for *DME-GFP* and *GFP-DME* knock-in, respectively. The horizontal lines indicate the positions of probes for Southern blotting. **c**–**f** Genotyping PCR and Southern blotting for individual T2 lines (**b**) and T3 plants (**e**) of *DME-GFP*, respectively. Arrow indicates the band of *DME-GFP* from gene targeting (see Supplementary Figure 3). Genotyping PCR and Southern blotting for individual T2 lines (**d**) and T3 plants (**f**) of *GFP-DME*, respectively. Arrow indicates the band of *GFP-DME* from gene targeting (see Supplementary Figure 3). **g** Analysis of seed abortion. Seeds from Col-0, homozygous *DME-GFP*, and *GFP-DME* knock-in T3 plants were analyzed. Scale bar, 1 mm. All PCR primers are as depicted in Supplementary Figure 3 and Supplementary Table 4

sequencing to analyze whether DNA methylation is affected in two independent homozygous T4 *ROS1-GFP* GT plants generated by our sequential GT strategy. We did not observe substantial changes in cytosine methylation in either the 5′ or 3′ homology arm regions (Supplementary Fig. 7), suggesting that our GT method did not affect the DNA methylation status of the targeted genomic locus.

## Discussion

Using our new approach for efficient and heritable GT in *Arabidopsis*, we achieved precise knock-ins, generating *ROS1-GFP*, *ROS1-Luc*, *DME-GFP*, and *GFP-DME* fusions, as well as gene replacements, generating *P1633A* and *F1648A* amino acid substitutions in DME. Only parental plant lines expressing Cas9 under the egg cell- and early embryo-specific promoter DD45 gave rise to efficient and heritable GT, without any need for a selection marker at the targeted locus. The fact that only DD45 promoter-driven Cas9 lines yielded heritable GT suggests that HDR may be more efficient in egg cells and/or early embryos than in other germline tissues (e.g., pollen and shoot apical meristem). We propose that germline GT occurs immediately after transformation, when *Agrobacteria* enter the Cas9-expressing ovule[63] to deliver the T-DNA containing sgRNA and donor DNA. Efficient HDR may occur in the egg cell and/or very early embryo, perhaps before T-DNA integration.

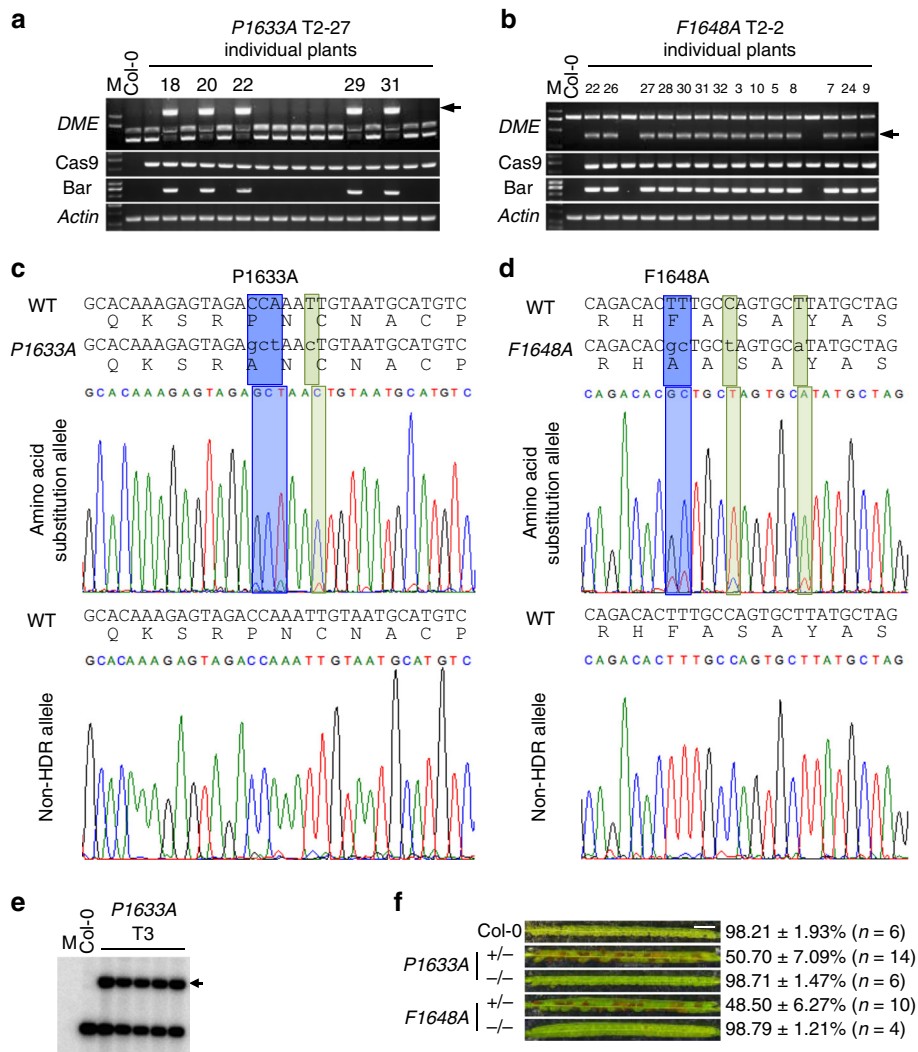

**Fig. 4** Single amino acid substitution at the endogenous *DME* locus by gene targeting. **a, b** Genotyping of T2 individual plants for *P1633A* and *F1648A*, respectively. Arrows indicate the specific sequence substitution as detected by PCR and restriction enzyme digestion. PCR primers are as depicted in Supplementary Figure 6a and Supplementary Table 4. Detailed information is described in Supplementary Figure 6. **c, d** Sequence confirmation of the *P1633A* and *F1648A* substitutions, respectively, in individual T2 plants. Blue highlights indicate amino acid substitution, green highlights indicate silent mutations. The sequence chromatograms are taken from FinchTV. **e** Southern blotting. The *P1633A* substitutions were confirmed by Southern blotting in heterozygous T3 plants. The arrow indicates the band caused by specific base substitution. **f** Analysis of seed abortion. Seeds from Col-0, heterozygous GT (+/−), and non-GT (−/−) *P1633A* and *F1648A* T3 plants were analyzed. Scale bar, 1 mm

### Table 2 Gene replacement efficiencies for the *DME* locus

| Construct | Parental line | T2 | | | | | | GT efficiency |
|---|---|---|---|---|---|---|---|---|
| | | Bulk | | Individual plants | | | | |
| | | PCR | Sequencing | Line name | PCR | Sequencing | | |
| *DME-P1633A* | DD45-#45 | 19/35 | 1/19 | T2-27 | 6/54 | 6/6 | | 5.3% (1/19) |
| *DME-F1648A* | DD45-#45 | 1/1 | 1/1 | T2-2 | 53/60 | 40/40 | | 100% (1/1) |

T2 bulk seedlings were genotyped by restriction digestion and PCR, and the positive samples were sequenced. For *DME-P1633A*, one of the 19 T2 bulk PCR positives had the P1633A amino acid substitution, and the rest had errors/mutations due to NHEJ. Fifty-four individual T2 plants from the positive line in T2 bulk screening were analyzed by restriction digestion and PCR, and six were found positive, and sequencing confirmed that all six had the P1633A substitution. For *DME-F1648A*, only one T2 population was obtained, which was found positive in the T2 bulk restriction digestion-PCR assay and sequencing showed that it was caused by F1648A substitution. Sixty T2 individual plants from this population were analyzed by restriction digestion-PCR and 53 were found positive. Forty of the 53 were tested by sequencing and all 40 were confirmed to have the F1648A substitution

Alternatively, HDR and the resulting GT may occur during the reproductive stage of T1 plants, when the T-DNA is already stably integrated. Five *GFP-DME* heterozygous T2 plants showed segregation from the Cas9 transgene (Fig. 3e, f), indicating that heritable knock-in occurred in T1 plants. The frequency of GT-positive plants in T2 populations ranged from 4/59 to 53/60 (Tables 1 and 2). The data are consistent with heritable GT events occurring in early embryos following the new transformation, in agreement with the strong activity of DD45 promoter in egg cells and early embryos[56].

All of the heritable GT events we observed were precise, without unexpected mutations or rearrangements at the target sites. The GT efficiency by our method was 5.3% for *DME P1633A* and was higher for other knock-ins or gene replacement (Tables 1 and 2). We analyzed T2 bulk DNA to determine whether the T-DNA copy numbers may contribute to efficient GT. Our results show that GT events were not related to T-DNA copy numbers of Cas9 or of the HDR donor transgene (Supplementary Fig. 8), suggesting that other unknown factors might be important. Additional research is required to understand and improve GT efficiency, and to apply this GT method to other plants including crops.

Here we revealed heritable GT and simple PCR-based identification, without the need of any selection marker at the target locus. This approach enables routine GT in *Arabidopsis*. Using egg cell- and early embryo-specific promoters to drive the expression of Cas9 or other site-specific nucleases, in combination with strategies for the effective delivery of donor DNA (such as described in ref. 4), might lead to efficient GT technologies in other plants, including crop plants.

## Methods

**Gene accession numbers**. *ROS1*, At2g36490; *DME*, At5g04560; *GL2*, At1g79840.

**Plant materials and growth condition**. The *Arabidopsis thaliana* accession Col-0 was used for all experiments. All plants were grown at 22 ˚C on half Murashige and Skoog (MS) medium with 1% sucrose or in soil with a 16 h light/8 h dark photoperiod. Parental T2 plants[53] were selected on the hygromycin (25 mg/L) containing MS plates for 10 days, then transplant in soil. The new transformation T1 lines were directly sowed in soil, and selected by three times Basta spray.

**Plasmid construction**. The optimized coding sequence of hSpCas9 (CRISPR/Cas9) plasmids for *GL2* GT, which were already reported[53], were constructed in pCambia1300. For all-in-one GT constructs, donor sequence was added to the published CRISPR/Cas9 constructs. For GT constructs for the sequential transformation strategy, AtU6 promoter-driven sgRNA and donor sequence were constructed in pCambia3301. All transformants were generated by the flower dipping method.

**DNA analysis**. Total DNA was extracted by the cethyltrimethyl ammonium bromide (CTAB) method from 10-day-old seedling for bulk analysis or 4- to 6-week-old for individual plant analysis. Extracted DNA was used for analysis of GT events by PCR and Southern blotting. Southern blotting was performed according to published protocols. Briefly, extracted DNA was digested overnight with chosen restriction enzymes, then separated on a 1.5% agarose gel, visualized by Image Lab Software and Gel Doc XR (BIO-RAD), and then transferred to nylon membrane (GE Healthcare). The probes were labeled with 32P-α-dCTP by using the Random primer DNA labeling kit (Takara). The hybridization signals were detected with a phosphor imager (Fuji). Un-cropped images of the most important Southern blots were supplied as Supplementary Fig. 10.

**RNA analysis**. For RT- and qRT-PCR, total RNA was extracted form 10-day-old or 4-week-old plants by using RNeasy Plant mini kit (Qiagen), treated with Turbo DNA-free (Ambion), and reverse transcribed by TransScript II (TransGen Biotech) with oligo (dT) primer. Then 1 μL of RT product was used as template for expression analysis. The raw data of some of the qPCR analysis are shown in Supplemental Fig. 9.

**Detection of GFP fluorescence and Luc luminescence**. GFP signal was observed in the roots of 3-day-old seedlings by confocal microscopy (Leica TCS SP8). Bright field and GFP fluorescence images were merged using ImageJ.

To determine firefly luciferase (Luc) reporter activity, 0.5 μM luciferin (Promega) in 0.01% Triton X-100 was sprayed onto 4-week-old mature leaves, followed by luminescence imaging using a high-performance CCD camera.

**DNA methylation analysis**. DNA methylation was analyzed by bisulfite sequencing. Total DNA was extracted using the CTAB method, and un-methylated cytosines were converted into uracil by using EZ DNA Methylation-Gold Kit (ZYMO RESEARCH). Genomic regions of interest were amplified by specific primers (Supplementary Table 4), then the amplicons were cloned into pMD-18 (Takara), and at least 27 independent colonies were sequenced. The sequence results were analyzed by Kismeth.

**Data availability**. The authors declare that all the data supporting the findings of this study are available within the paper and its supplementary information files. The data sets generated or analyzed during the current study are available from the corresponding author on reasonable request. We deposited our DD45::CRSIPR/ Cas9 parental lines, DD45-#58 and DD45-#70, to the Arabidopsis Biological Resource Center (ABRC). The seeds were assigned the stock numbers CS69955 and CS69956, respectively. The two homozygous DD45::CRISPR/Cas9 parental lines could retain a high rate of GT when they are propagated to future generations with hygromycin selection.

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

## Acknowledgements

This work was supported by the Chinese Academy of Sciences to J.-K.Z., and by Grant for Basic Science Research Projects from The Sumitomo Foundation to D.M. We would like to thank Life Science Editors for editorial assistance, Ms. Wencan Zhang for assistance, the Plant Cell Biology Core Facility at the Shanghai Center for Plant Stress Biology for assistance with confocal microscopy.

## Author contributions

D.M., W.X.Z., and J.-K.Z. designed the research; W.X.Z. and D.M. performed the experiments with assistance from W.J.Z., and Z.F.; D.M. and J.-K.Z. supervised the project; D.M., W.X.Z. and J.-K.Z. wrote the paper.

## Additional information

**Competing interests:** The authors declare no competing interests.

