## [Peer Review File · Nature Communications]

Reviewers' comments:

Reviewer #1 (Remarks to the Author):

I read the manuscript of Miki et al with great interest. Reading the abstract, I got the impression that the author supplied the community with a simple protocol that on the spot will enable every Arabidopsis laboratory to perform gene targeting experiments with high efficiency. However, having a closer look at the text it turns out that the authors indeed took a heroic effort including many fruitless attempts and developed a new kind of two step strategy for gene targeting in Arabidopsis which in the end is novel but also quite time consuming. So in respect of time the procedure is by far not as efficient as by the impressive frequencies.

In any case, I don't think that in its present form the manuscript is publishable, as important information is missing for a critical evaluation of the produce. The authors report that with two Cas9 lines driven by the egg-cell-specific promoter, DD45-#58 and #70, they were able to obtain efficient targeting. But how many lines did they test in all? Taking the numbering into account if they tested a hundred lines and if other labs have to go through this procedure the gene targeting frequency in terms of work load would not be 6 to 8% but 0.06 to 0.08% which would indeed not differ from the gene targeting frequencies reported before. I think this point has to be clarified by the authors, to give the readers the chance to fairly evaluate the prospects using the technology. The authors have to show GT data of at least ten randomly chosen lines where Cas9 is under egg-cell specific expression.

Moreover, if it really is so hard to find lines with efficient targeting, it will be almost impossible to reproduce these frequencies with newly generated lines. Therefore, the authors would have to freely distribute these two lines in the community and they should state that they will do this on request freely. I also ask myself whether it is possible to obtain targeted mutations without Cas9 transgene from these lines.

I regard title and abstract as misleading. The authors try to sail under a false flag. Title as well as abstract have to reveal that a two-step strategy was developed to achieve the high GT frequencies. Moreover, supplemental figure 2 has to be integrated into the main text to help to understand the reader the (quite complex and time consuming) strategy.

Reviewer #2 (Remarks to the Author):

The precise induction of the DNA double-stranded breaks (DSBs) at the targeted locus and

proper delivery of the repair templates at the timing of DSBs induction is critical for the establishment of an efficient gene targeting (GT) system in plants.

Here authors used a “sequential transformation method” to achieve an efficient GT in Arabidopsis. In this method, Cas9 protein has been already expressing in host plants under the control of a germline-specific promoter DD45, and repair templates and gRNA expression construct were delivered by the second in planta transformation.

Although the methods presented here seemed to be repeatable and could apply to knock-in, and targeted induction of the substitutions at a targeted locus, some of the important information is missing from this manuscript.

The high efficient GT could be established by the sequential transformation method. However, sequential transformation method has been already reported in rice GT experiment (Endo et al. 2016). Authors have to cite reference papers properly.

It is not clear whether authors have utilized bialaphos or Basta selection before confirming GT by PCR. Figure 1c indicates that bar genes are detected in all lines.

So, there is a possibility GT can be done via homologous recombination between the targeted locus and pre-integrated repair templates. Furthermore, in summary, authors wrote that authors did not use any selection marker. However, if authors utilize bar selection, this selection process do apparently concentrate GT candidate plants that contained repair templates.

I am not sure, the GT efficiency shown in Table 1 can indicate a practical GT efficiency properly. For example, 7.7% (2/26) means GT events occurred in two T2 bulk derived from 26 T1 lines.

However, this means 2 T1 plants out of 26 T1 plants contained somatic cells which have succeeded in GT, and GT plants were produced in 7/56 or 4/59 of the T2 plants. So, I think a practical GT efficiency should be estimated as 2/26 by 7/56.

Although authors wrote that “ The epigenetic status of gene-targeted endogenous loci has not been reported in plants”, Lieberman-Lazarovich et al.(2013) have reported epigenetic alterations at genomic loci modified by gene targeting in Arabidopsis. Authors should cite this paper and discuss properly.

The method presented here should be useful for the knock-in experiment in Arabidopsis. However, at present, in planta transformation method can be applied exclusively in Arabidopsis. So authors should discuss limitation and future application of this method for the establishment of a general strategy of GT in plants.

Minor comments:

Line 30

Authors used the technical term “short guide RNA-targeted gene knock-in.” However, usually, the term “single guide RNA (sgRNA) targeted” has been used for the CRISPR/Cas9 mediated induction of DSBs.

Furthermore, short RNA can also be used as a repair template for gene targeting. The term “short guide RNA-targeted gene knock-in” could mislead authors.

Reviewer #3 (Remarks to the Author):

The ms by Miki et al claims the development of a method for the efficient and selection-marker free gene targeting (GT) in *Arabidopsis thaliana*. Development of such method is direly needed in the plant community. Although many papers have described GT in plants, until now only with low frequency. Therefore, the development of such method would be of the uttermost importance and be most wellcome in the field. After reading the current ms I have to conclude that the authors have made an important step in the direction of such methodology. The authors show convincingly that by using a specific promoter to drive the expression of the Cas protein in the developing embryo, re-transformation of such embryos can lead to GT with high efficiency of the genes that were tested. However, the procedure does not work in all cases and it is not clear why this is the case. Nevertheless, the procedure described forms an important step forward that can be used in the future by the current authors and others for futher improvement.

Detailed comments:

Title claims “selection-marker free gene targeting”. Also reiterated in the discussion in line 202. I see that in more publications, but I find this misleading. The title refers to the fact that the GT event itself is not selected. Nevertheless, selection markers are used to select for transformants, which include GT events. Therefore, selection markers are still essential in the procedure.

Line 41-43 This sentence requires some reformulation. Precise sequence changes only occur in GT procedures, when a homologous substrate is offered. Otherwise a DSB is restored by HR using the sister chromatid as template leading to precise restoration of the original sequence.

Line 47 The first GT event in plants showing for the first time restoration of an endogenous gene at its locus by an incoming DNA molecule was published by Offringa et al. , EMBO J 9 (1990)3077–3084. Ref8 only showed that HR between incoming DNA and an endogenopus gene is possible without showing where the recombined DNA molecule integrated in the genome.

Line 51 The use of positive-negative selection schemes has been tested in other plants such as tobacco, but was not successful. See for instance: Morton and Hooykaas, *Mol Breeding* 1 (1995)123-132.

Line 52 As a reference for the use of specific nucleases to increase GT I would like to point to the article by Choulika et al, *Mol Cell Biol.* 15(1995)1968–1973, which was the first to use such nuclease (Sce1 in this case) in a heterologous system to increase GT events 100 fold.

Line 107-110 Were the T1 plants from which the T2 seedlings were analyzed for GT also checked by PCR? As some of the T2 plants were homozygous for GT, this suggests that the T1 parent may have been a GT plant already. Alternatively, the GT event took place during embryo formation of the T2 seeds and was the GT event copied into the homologous chromosome during DSB repair of DSBs due to ongoing CRISPR activity.

Line 107-110 and similarly for the other GT cases. As only about 10% of the re-transformed T1 plants showed GT events, it would be extremely important to find out if there is something special in these lines. Is the second T-DNA present in high copy number, expressed highly or integrated close-by the target locus? In the analysis of the GT T2 plants in Fig1 PCR shows that both T-DNAs (the one expressing Cas and the second with the sgRNA and the donor DNA) are still present in all the T2 seedlings. Were these T2 plants first selected on both hygromycin and ppt? If not, why is no segregation seen of these T-DNAs? In Figs 2 and 3 the same is the case for the donor T-DNA and the Cas T-DNA, respectively. This is the more remarkable as the same Cas T-DNA is present in all the T2 lines. Was the starting material in experiment 1 and 3 homozygous for the Cas T-DNA and in experiment 2 hemizygous?

Line 133 The GFP targeting events were only found in the Cas9 /58 background. In this case the Cas9/70 background apparently did not generate GT events. Why was this the case? Was line /70 not used or not enough T2 lines obtained or does this show that the procedure is not fully reproducible? Similarly for the DME-GFP targeting described in line 153.

Line 191 The epigenetic status of GT loci has in fact been studied in plants by Lieberman-Lazarovich et al., *PLoS ONE* 8(2013) e85383: Epigenetic Alterations at Genomic Loci Modified by Gene Targeting in *Arabidopsis thaliana*. These authors found either no change, loss of methylation or unstable methylation over generations. It is remarkable that the authors did not find any change in methylation in any of their GT lines. How many different GT lines were analyzed in total?

Materials and Methods should include a short description of the Cas9 gene used. Was it codon-optimized for *Arabidopsis*? For how long were the T1 plants first grown on hygromycin or ppt before molecular analysis. In pools of T2 plants, how many plants were pooled together? Lines

with the highest CRISPR efficiency were selected (line 100). How was this done and from how many starting lines? These lines were used as parental lines (line 101). Were these first grown on hygromycin or were lines homozygous for the construct first selected?

Response to the Reviewers' comments:

Reviewer #1

In any case, I don't think that in its present form the manuscript is publishable, as important information is missing for a critical evaluation of the produce. The authors report that with two Cas9 lines driven by the egg-cell-specific promote, DD45-#58 and #70, they were able to obtain efficient targeting. But how many line did they test in all? Taking the numbering into account if they tested a hundred lines and if other lab have to go through this procedure the gene targeting frequency in times of work load would not be 6 to 8% but 0.06 to 0.08% which would indeed not different from the gene targeting frequencies reported before. I think this point has to be clarified by the authors, to give the readers the chance to fairly evaluate the prospects using the technology. The authors have to show GT data of at least ten randomly chosen lines where Cas9 is under egg-cell specific expression.

Response: In this research, we tested just two independent CRISPR/Cas9 parental lines for each germline-specific promoter, namely the two that showed the highest mutation efficiency among 32-36 lines for every promoter construct according to our previous report (Mao et al, 2016, Plant Biotechnol. J.)⁵³. We obtained GT-positive plants from both DD45 promoter background parental lines but not from any of the other promoter

background parental lines. The fact that efficient GT was obtained from two independent lines of the DD45 promoter parental background suggests that the efficient GT was not due to something special in a particular line.

Moreover, if it really so hard to find lines with efficient targeting, it will be almost impossible to reproduce these frequencies. with newly generated lines. Therefore, the authors would have to freely distribute these two lines in the community and they should state that they will do this on request freely.

Response: Once our manuscript is accepted, we will deposit the two DD45::CRSIPR/Cas9 parental lines to ABRC to distribute freely to the community. We now state this in the revised manuscript.

I also ask myself whether it is possible to obtain targeted mutations without Cas9 transgene from these lines.

Response: Yes, it is possible. As shown Fig. 2d and 2f, we obtained lines which are with GT and without the Cas9 transgene.

I regard tile and abstract as misleading. The authors try to sail under a false flag. Title as well as abstract have to reveal that a two-step strategy was developed to achieve the high GT frequencies.

Response: We have changed the tile to “Efficient CRISPR/Cas9-mediated gene targeting in *Arabidopsis* using sequential transformation”.

The Summary was also changed as follows.

Line 31: Here we report a simple and efficient method for gene targeting in *Arabidopsis* using sequential transformation.

Moreover, supplemental figure 2 has to be integrated into the main text to help to understand the reader the (quite complex and time consuming) strategy.

Response: We added a brief explanation of the sequential transformation strategy

depicted in Extended Data Figure 2 as follows:

Line 105-114: We used these Cas9-expressing plants as parental lines for new transformations with a construct containing: (i) HDR donor sequences, (ii) sgRNAs targeting a genomic locus of interest, (iii) a selectable marker for plants that are positive for the donor construct (Extended Data Figure 2, Fig. 1a). The new transformation T1 transgenic plants were selected using the Basta resistance gene. These T1 plants express Cas9 and a specific sgRNA, and contain a specific HDR donor sequence. T1 seeds were harvested and germinated without selection on MS plates; 20-30 of the resulting T2 seedlings were subsequently pooled together, and GT events were analyzed by PCR in bulk. Further, another batch of T2 plants from the bulk positive lines were investigated as individual plants (Extended Data Figure 2).

Reviewer #2

The high efficient GT could be established by the sequential transformation method. However, sequential transformation method has been already reported in rice GT experiment (Endo et al. 2016). Authors have to cite reference papers properly.

Response: We thank the reviewer for pointing this out and have added two references as follows.

Line 100-102: Next, we used a “sequential transformation method” to evaluate GT efficiency^{35,41} in parental *Arabidopsis* plants that already express Cas9 from a germline-specific (DD45, Lat52, YAO or CDC45) promoter (Extended Data Figure 2).

It is not clear whether authors have utilized bialaphos or Basta selection before confirming GT by PCR. Figure 1c indicates that bar genes are detected in all lines.

Response: We apologize for the confusion. We selected T1 plants based on Basta resistance, but did not use bialaphos or Basta to select T2 and their future generations. We detected GT in T2 by PCR. We clarified the lack of antibiotic selection in T2 in Extended Data Figure 2.

So, there is a possibility GT can be done via homologous recombination between the targeted locus and pre-integrated repair templates.

Response: Yes, there is such a possibility, which is now discussed in lines 219-224.

Furthermore, in summary, authors wrote that authors did not use any selection marker. However, if authors utilize bar selection, this selection process do apparently concentrate GT candidate plants that contained repair templates.

Response: The bar selection marker was used to select for T1 transformants containing the repair template. We used “selection marker free” in reference to the GT event itself, which was not based on antibiotic nor herbicide resistance. However, we appreciate this concern and to avoid confusion we changed the title and text as follows.

Title: “Efficient CRISPR/Cas9-mediated gene targeting in *Arabidopsis* using sequential transformation”

Text: selection marker-free at the targeted locus

I am not sure, the GT efficiency shown in Table 1 can indicate a practical GT efficiency properly. For example, 7.7% (2/26) means GT events occurred in two T2 bulk derived from 26 T1 lines.

However, this means 2 T1 plants out of 26 T1 plants contained somatic cells which have succeeded in GT, and GT plants were produced in 7/56 or 4/59 of the T2 plants. So, I think a practical GT efficiency should be estimated as 2/26 by 7/56.

Response: We appreciate the comment, and have tried to make it clear that our GT frequency was calculated based on the number of T2 populations examined.

Although authors wrote that “The epigenetic status of gene-targeted endogenous loci has not been reported in plants”, Lieberman-Lazarovich et al.(2013) have reported epigenetic alterations at genomic loci modified by gene targeting in *Arabidopsis*. Authors should cite this paper and discuss properly.

Response: Thank you. We added the reference and rewrote the text as follows.

Line 204-210: ZFN-mediated gene targeting of the endogenous locus PPOX in plants reportedly alters its epigenetic status⁶². We performed individual locus bisulfite

sequencing to analyze whether DNA methylation is affected in two independent homozygous T4 *ROSI-GFP* GT plants generated by our sequential GT strategy. We did not observe substantial changes in cytosine methylation in either the 5' or 3' homology arm regions (Extended Data Figure 8), suggesting that our GT method did not affect the DNA methylation status of the targeted genomic locus.

The method presented here should be useful for the knock-in experiment in Arabidopsis. However, at present, in planta transformation method can be applied exclusively in Arabidopsis. So authors should discuss limitation and future application of this method for the establishment of a general strategy of GT in plants.

Response: We agree that our GT method is currently limited to a few plant species where in planta transformation is successful. Our results suggest that expression of CRISPR/Cas9 in the ovule and early embryo is important for efficient GT. In the future, efficient GT technology can be established based on this principle, including in crop plants, as we discussed in lines 239-244.

Minor comments:

Line 30

Authors used the technical term “short guide RNA-targeted gene knock-in.” However, usually, the term “single guide RNA (sgRNA) targeted” has been used for the CRISPR/Cas9 mediated induction of DSBs.

Furthermore, short RNA can also be used as a repair template for gene targeting. The term “short guide RNA-targeted gene knock-in” could mislead authors.

Response: We changed “short guide RNA” to “single-guide RNA” (line 32).

Reviewer #3

Title claims “selection-marker free gene targeting”. Also reiterated in the discussion in line 202. I see that in more publications, but I find this misleading. The title refers to the fact that the GT event itself is not selected. Nevertheless, selection markers are used to select for transformants, which include GT events. Therefore, selection markers are still

essential in the procedure.

Response: We appreciate this point and to avoid confusion we have changed the title and text as follows.

Title: “Efficient CRISPR/Cas9-mediated gene targeting in *Arabidopsis* using sequential transformation”

Text: selection marker-free at the targeted locus

Line 41-43 This sentence requires some reformulation. Precise sequence changes only occur in GT procedures, when a homologous substrate is offered. Otherwise a DSB is restored by HR using the sister chromatid as template leading to precise restoration of the original sequence.

Response: Thank you. We rewrote the sentence as follows.

Line 44-47: Repair of these DSBs via error-prone non-homologous end-joining (NHEJ) leads to random mutations, whereas error-free homology-directed repair (HDR) creates precise sequence changes when a homologous DNA substrate is provided.

Line 47 The first GT event in plants showing for the first time restoration of an endogenous gene at its locus by an incoming DNA molecule was published by Offringa et al. , EMBO J 9 (1990)3077–3084. Ref8 only showed that HR between incoming DNA and an endogenous gene is possible without showing where the recombined DNA molecule integrated in the genome.

Response: We added the Offringa et al., (1990) reference.

Line 51 The use of positive-negative selection schemes has been tested in other plants such as tobacco, but was not successful. See for instance: Morton and Hooykaas, Mol Breeding 1 (1995)123-132.

Response: We added the Morton and Hooykaas, (1995) reference.

Line 52 As a reference for the use of specific nucleases to increase GT I would like to

point to the article by Choulika et al, Mol Cell Biol. 15(1995)1968–1973, which was the first to use such nuclease (Sce1 in this case) in a heterologous system to increase GT events 100 fold.

Response: We added the Choulika et al., (1995) reference.

Line 107-110 Were the T1 plants from which the T2 seedlings were analyzed for GT also checked by PCR? As some of the T2 plants were homozygous for GT, this suggests that the T1 parent may have been a GT plant already. Alternatively, the GT event took place during embryo formation of the T2 seeds and was the GT event copied into the homologous chromosome during DSB repair of DSBs due to ongoing CRISPR activity.

Response: We did not check the T1 plants for GT by PCR. We agree with the reviewer that for the T2 plants homozygous for GT their T1 parent may have been a GT plant already, or the GT event occurred later during embryo formation of the T2 seeds. We hope to investigate this in the future.

Line 107-110 and similarly for the other GT cases. As only about 10% of the re-transformed T1 plants showed GT events, it would be extremely important to find out if there is something special in these lines. Is the second T-DNA present in high copy number, expressed highly or integrated close-by the target locus?

Response: Thank you for this suggestion. We now analyzed the copy numbers of the re-transformed donor and parental Cas9 transgenes in the T2 bulk population by qPCR, and the results are added as Extended Data Figure 9. These results indicate that the donor and Cas9 copy numbers do not contribute to efficient GT, suggesting that other unknown factors are important. We hope to investigate this in future.

We added new sentences to line 233-238 as follows:

Line 233-238: We analyzed T2 bulk DNA to determine whether the T-DNA copy numbers may contribute to efficient GT. Our results show that GT events were not related to T-DNA copy numbers of Cas9 or of the HDR donor transgene (Extended Data Figure 9), suggesting that other unknown factors might be important. Additional research is required to understand and improve GT efficiency, and to apply this GT

method to other plants including crops.

In the analysis of the GT T2 plants in Fig1 PCR shows that both T-DNAs (the one expressing Cas and the second with the sgRNA and the donor DNA) are still present in all the T2 seedlings. Were these T2 plants first selected on both hygromycin and ppt?

Response: We did not perform any selection for T2 and the future generations. We now clarified the lack of hygromycin and basta selections in T2 in Extended Data Figure 2.

If not, why is no segregation seen of these T-DNAs? In Figs 2 and 3 the same is the case for the donor T-DNA and the Cas T-DNA, respectively. This is the more remarkable as the same Cas T-DNA is present in all the T2 lines. Was the starting material in experiment 1 and 3 homozygous for the Cas T-DNA and in experiment 2 hemizygous?

Response: The parental lines were a mixed population, which were selected via hygromycin treatment (Methods); Majority in the population should be homozygous for the CRISPR/Cas9 transgene, and some are hemizygous plants. In their progenies, some segregated away from the transgenes, and some did not. We hope to investigate this in the future.

Line 133 The GFP targeting events were only found in the Cas9 /58 background. In this case the Cas9/70 background apparently did not generate GT events. Why was this the case? Was line /70 not used or not enough T2 lines obtained or does this show that the procedure is not fully reproducible? Similarly for the DME-GFP targeting described in line 153.

Response: We did not use the #70 line for these experiments because the seeds from this parental line were limiting at the time.

Line 191 The epigenetic status of GT loci has in fact been studied in plants by Lieberman-Lazarovich et al., PLoS ONE 8(2013) e85383: Epigenetic Alterations at Genomic Loci Modified by Gene Targeting in Arabidopsis thaliana. These authors found either no change, loss of methylation or unstable methylation over generations. It is

remarkable that the authors did not find any change in methylation in any of their GT lines. How many different GT lines were analyzed in total?

Response: We analyzed the DNA methylation status in two independent T4 lines (Extended Data Figure 8). We added the reference and rewrote the text as follows.

Line 204-210: ZFN-mediated gene targeting of the endogenous locus PPOX in plants reportedly alters its epigenetic status⁶². We performed individual locus bisulfite sequencing to analyze whether DNA methylation is affected in two independent homozygous T4 ROS1-GFP GT plants generated by our sequential GT strategy. We did not observe substantial changes in cytosine methylation in either the 5' or 3' homology arm regions (Extended Data Figure 8), suggesting that our GT method did not affect the DNA methylation status of the targeted genomic locus.

Materials and Methods should include a short description of the Cas9 gene used. Was it codon-optimized for Arabidopsis? For how long were the T1 plants first grown on hygromycin or ppt before molecular analysis.

Response: We used the optimized coding sequence of hSpCas9 as reported in our previous paper (Mao et al, 2016, Plant Biotechnol. J.)⁵³. We rewrote the Methods section as follows.

Line 264-273: Plant materials and growth condition

The *Arabidopsis thaliana* accession Col-0 was used for all experiments. All plants were grown at 22 °C on half Murashige and Skoog (MS) medium with 1% sucrose or in soil with a 16 hour light/8 hour dark photoperiod. Parental T2 plants⁵³ were selected on the hygromycin (25mg/L) containing MS plates for 10 days, then transplanted in soil. The new transformation T1 lines were directly sowed in soil, and selected by three times Basta spray.

Plasmid construction

The optimized coding sequence of hSpCas9 (CRISPR/Cas9) plasmids for *GL2* gene targeting, which were already reported⁵³, were constructed in pCambia1300.

In pools of T2 plants, how many plants were pooled together?

Response: 20-30 T2 seedlings were pooled together. We added this information in Extended Data Figure 2.

Lines with the highest CRISPR efficiency were selected (line 100). How was this done and from how many starting lines?

Response: We analyzed 32-36 independent T1 lines to select two promoter-Cas9 parental lines. We chose two CRISPR lines with the highest mutation rates at the GL2 locus (Mao et al, 2016, Plant Biotechnol. J.)⁵³.

These lines were used as parental lines (line 101). Were these first grown on hygromycin or were lines homozygous for the construct first selected?

Response: Parental T2 lines were selected on hygromycin-containing MS plates (25mg/L) for 10 days, and then transplanted in soil (Methods and Extended Data Figure 2). As mentioned above, we did not have a detailed characterization of those T2 lines. Based on our GT results, we think that the parental lines are a heterogeneous population of homozygous (majority) and hemizygous plants.

Reviewers' Comments:

Reviewer #1 (Remarks to the Author):

The revised version of Liu et al was modified in part according to the suggestions of the reviewers. Unfortunately, I am not satisfied with all changes and due to some clarifications of the authors I have now much more severe doubts than before that the procedure is indeed an improvement for the field

Obliviously, the authors developed a novel two-step procedure that is indeed different to all other approaches taken before to solve the gene targeting problem in plants. However, reading the revision and the response letter now and only now it became clear to me that that in contrast to what is claimed in the tile the procedure is not so efficient and it definitely works not with higher frequencies as other protocols published before. The frequencies given by the authors in the tables are misleading and give the impression of an extremely efficient procedure. As reviewer 2 states in his comment one has to take into account the bulk screened as well as the individual screened T2 plants of the respective experiments. Thus, as stated by him/her the frequencies is not as claimed 2/26 but 2/26 by 7/56. I have to admit ,that when I wrote my first review I was so preoccupied by working out the fact that the authors applied a two-step strategy, that I did not question the presentation of the table and was misled given by the authors referring only to T2 numbers. The reported targeting frequency are at maximum is in the percent range Similar or even higher numbers have already been achieved by the Voytas group years ago with the geminiviral approach as well as by recent reports using Cpf1.

Further points:

- As the two-step protocol is difficult to understand I really think inclusion of the supplementary figure 2 on the details of the protocol is essential (I appreciated the improvements to the figure made in the revision) and will definitely help the reader.
- I think that authors have indeed to state not only in the letter but also in the text that they took the two best expressers of 32-36 lines available for their experiments.
- In contrast, to what they claim in the letter I could not find any sentence the text that the two Cas9 expression lines are freely available to the community. To include such a sentence is essential.
- The percentage numbers given in table 1 are not correct. The authors have to calculate frequency taking the bulk as well as the individual plants into account. So for DD45-#58 the GT

efficiency not 7,7 % but 0.96 %, ect.

Reviewer #2 (Remarks to the Author):

Here are my comments on the revised manuscript.

Lines 36-38: Although authors wrote that “Our approach enables routine, fine manipulation of the Arabidopsis genome and will aid in the development of effective gene targeting tool in other plants”, in planta transformation technique could not be applicable on other plants – combination of the ovule-specific promoter driven Cas9 expression and in planta transformation is not a universal technology.

Lines 218-221: If GT occurs immediately after transformation in ovules, the authors should get non-chimeric genome editing plants in a T1 generation. However, authors did not check T1 generation plants whether they can get non-chimeric genome editing plants.

Reviewer #3 (Remarks to the Author):

The ms by Miki et al describes the development of a sequential transformation protocol with which GT events can be obtained effectively in the T2 population of selected lines. In the first step plant lines are selected containing a highly efficient CRISPR/Cas nuclease providing high frequency mutagenesis. Subsequently a donor fragment is introduced in these lines which now is used efficiently for repair of the DSB site made by the nuclease.

I think this approach, although laborious, will be of great interest to the plant community.

The authors have dealt with the critical points which I raised in the original submission in an adequate way. In my opinion the current version is acceptable for publication. Paul Hooykaas

Response to the Reviewers' comments:

Reviewer #1

The frequencies given by the authors in the tables are misleading and give the impression of an extremely efficient procedure. As reviewer 2 states in his comment one has to take into account the bulk screened as well as the individual screened T2 plants of the respective experiments. Thus, as stated by him/her the frequencies is not as claimed 2/26 but 2/26 by 7/56.

Response: We appreciate the comment, and have tried to make it clear that our GT frequency was calculated based on the number of T2 populations examined.

Similar or even higher numbers have already been achieved by the Voytas group years ago with the geminiviral approach as well as by recent reports using Cpf1.

Response: Prof. Voytas's group published more than 10 papers for GT in various plant species. Especially high efficiency GT frequency, ~11%, was reported in tomato by using the combination of geminivirus and CRISPR/Cas9 (Cermak et al., 2015) (GT events were selected based on antibiotic resistance). Thus, according to reviewers' and editor's

suggestions, we minimized the use of adjective words. The word “efficient” is now removed from our title, and the Abstract has been re-written.

As the two-step protocol is difficult to understand I really think inclusion of the supplementary figure 2 on the details of the protocol is essential (I appreciated the improvements to the figure made in the revision) and will definitely help the reader.

Response: Thank you very much for your suggestion. We moved Supplementary Figure 2 into main text as Figure 1.

I think that authors have indeed to state not only in the letter but also in the text that they took the two best expressers of 32-36 lines available for their experiments.

Response: We modified the main text to describe the 32-36 parental lines as follows.

Lines 145-147: We used the two highest efficiency CRISPR/Cas9 lines, which were screened from 32 to 36 independent T1 lines based on the mutation rates at the *GL2* locus, for each specific promoter⁵³.

In contrast, to what they claim in the letter I could not find any sentence the text that the two Cas9 expression lines are freely available to the community. To include such a sentence is essential.

Response: We added the following sentence in Data availability in Methods section.

“We deposited our DD45::CRISPR/Cas9 parental lines, DD45-#58 and DD45-#70, to the Arabidopsis Biological Resource Center (ABRC). The seeds were assigned the stock numbers CS69955 and CS69956, respectively.”

And we also added a sentence at the end of manuscript as Note and the cover page of Supplementary Information.

“The DD45::CRSIPR/Cas9 parental lines, DD45-#58 and DD45-#70, are available at the Arabidopsis Biological Resource Center (ABRC), with the stock numbers CS69955 and

CS6956, respectively.”

The percentage numbers given in table 1 are not correct. The authors have to calculate frequency taking the bulk as well as the individual plants into account. So for DD45-#58 the GT efficiency not 7,7 % but 0.96 %, ect.

Response: As described above, our GT efficiency was calculated based on the number of T2 populations examined. Further, we think this comment is linked to the question when are the GT events established. In the future, we hope to investigate when our GT is established and its frequency.

Reviewer #2

Lines 36-38: Although authors wrote that “Our approach enables routine, fine manipulation of the Arabidopsis genome and will aid in the development of effective gene targeting tool in other plants”, in planta transformation technique could not be applicable on other plants – combination of the ovule-specific promoter driven Cas9 expression and in planta transformation is not a universal technology.

Response: We thank the reviewer and editor for pointing this out. We agree that our GT method is currently limited to a few plant species where in planta transformation is successful. We thus deleted this sentence from the Abstract.

Lines 218-221: If GT occurs immediately after transformation in ovules, the authors should get non-chimeric genome editing plants in a T1 generation. However, authors did not check T1 generation plants whether they can get non-chimeric genome editing plants.

Response: We did not check the T1 plants for GT. It is still not clear when our GT is established. Thus the occurrence of GT immediately after transformation in ovules is only a hypothetical working model at this time. We hope to investigate this in the future.

Reviewer #3

Dear Prof. Paul Hooykaas,

We really appreciate all of your comments.